# On Efficiency in Hierarchical Reinforcement Learning

**Zheng Wen**
DeepMind
zhengwen@google.com

**Doina Precup**
DeepMind
doinap@google.com

**Morteza Ibrahimi**
DeepMind
mibrahimi@google.com

**Andre Barreto**
DeepMind
andrebarreto@google.com

**Benjamin Van Roy**
DeepMind
benvanroy@google.com

**Satinder Singh**
DeepMind
baveja@google.com

## Abstract

Hierarchical Reinforcement Learning (HRL) approaches promise to provide more efficient solutions to sequential decision making problems, both in terms of statistical as well as computational efficiency. While this has been demonstrated empirically over time in a variety of tasks, theoretical results quantifying the benefits of such methods are still few and far between. In this paper, we discuss the kind of structure in a Markov decision process which gives rise to efficient HRL methods. Specifically, we formalize the intuition that HRL can exploit well repeating "subMDPs", with similar reward and transition structure. We show that, under reasonable assumptions, a model-based Thompson sampling-style HRL algorithm that exploits this structure is statistically efficient, as established through a finite-time regret bound. We also establish conditions under which planning with structure-induced options is near-optimal and computationally efficient.

## 1   Introduction

Hierarchical reinforcement learning (HRL) refers to the ability of an agent to act and plan at multiple levels of temporal abstraction [Sutton et al., 1999, Barto and Mahadevan, 2003]. In principle, this ability can present several benefits: (1) more efficient exploration, by employing policies that help an agent circulate more efficiently over the state space; (2) more efficient credit assignment, because temporally extended models propagate credit over many time steps, and the same stream of data can be re-used to learn about many possible contingencies (for example, expressed as sub-goals); and (3) the ability to solve smaller problems and compose the resulting policies and models quickly in new situations. From a theoretical point of view, (1) has been investigated in recent work [Fruit et al., 2017]. Aspects (2) and (3) have received empirical validation in a large number of papers, but not much theoretical analysis, except for some special cases, such as the work of Mann et al. [2015]. In this paper, we present two general results which highlight the types of problems in which HRL is expected to provide benefits, in terms of planning speed, as well as in terms of statistical efficiency. First, as has been highlighted empirically in the past, having repeated structure in the Markov decision process (MDP) can lead to large speedups in both aspects. Second, in terms of planning, HRL provides benefits when we are able to "insulate" well sub-problems that can be solved in isolation, and whose solutions can then be "stitched" together. We formalize the latter intuition.

**Contributions:** First, we formalize a notion of MDP decomposition into sub-problems, using state partitions. Second, we show that the existence of hierarchical structure in the environment can lead to statistically efficient learning, by extending the results in Osband et al. [2013] to establish a regret bound that separates errors due to sub-optimal planning and errors due to learning. If the original

MDP can be decomposed into repeating problems that are relatively small, the expected regret of a posterior sampling exploration algorithm can be dramatically reduced. Finally, we study planning under decompositions of the original problem. We establish a relationship between the complexity of planning and the number of contingencies under which sub-problems are solved, formalized through a notion of *exit profiles*. We show formally that near-optimal planning can be obtained much faster if the original problem can be partitioned into repeated problems that are small and well separated.

## 2 Problem formulation

Consider a finite-time horizon MDP $\mathcal{M} = \langle \mathcal{S}, \mathcal{A}, P, r, s_e, s_0 \rangle$, where $\mathcal{S}$ is a finite state space, $\mathcal{A}$ is a finite action space, $P$ and $r$ respectively encode the transition model and the reward model, $s_e \in \mathcal{S}$ is a fixed terminal state, and $s_0 \in \mathcal{S}$ is a fixed initial state[1]. If the agent takes action $a \in \mathcal{A}$ at a non-terminal state $s \in \mathcal{S} \setminus \{s_e\}$, it receives a random reward drawn from distribution $r(\cdot|s, a)$, and transits to the next state $s' \in \mathcal{S}$ with probability $P(s'|s, a)$. Without loss of generality, we assume that the support of reward distribution $r(\cdot|s, a) \subseteq [0, 1], \forall s \in \mathcal{S} \setminus \{s_e\}, \forall a \in \mathcal{A}$. We use $\bar{r}(s, a)$ to denote the mean of $r(\cdot|s, a)$. In this paper, we consider two different but related problems in MDPs: planning and reinforcement learning (RL).

In the planning setting, the agent knows $\mathcal{M}$, and its goal is to compute a near-optimal policy, $\pi : \mathcal{S} \to \mathcal{A}$, i.e., a policy that maximizes the expected total reward: $\max_\pi \mathbb{E}_\pi \left[ \sum_{h=1}^{\tau-1} r_h \right]$, where $r_h \sim r(\cdot|s_h, a_h)$ is the reward received by taking action $a_h = \pi(s_h)$ in time period $h$, and $\tau$ is a random variable that denotes the time at which the agent enters $s_e$. To simplify the exposition, we assume that under any policy $\pi$, $\tau \leq \tau_{\max}$ with probability 1 and $\mathbb{E}[\tau] \leq H$.

In the RL setting, the agent knows $\mathcal{S}$, $\mathcal{A}$, $s_e$, $s_0$, but does not know $P$ or $r$. The agent will repeatedly interact with $\mathcal{M}$ for $T$ episodes. An episode always starts at $s_0$ and ends immediately upon entering $s_e$. The agent's goal is to maximize its expected cumulative reward over the $T$ episodes, $\max \sum_{t=1}^{T} \mathbb{E} \left[ \sum_{h=1}^{\tau_t} r_{th} \right]$, where we use $t$ to index episodes and $h$ to index periods in an episode, $r_{th} \sim r(\cdot|s_{th}, a_{th})$ is the reward received in period $h$ of episode $t$ and $\tau_t$ is the duration of episode $t$. Note that an RL algorithm might call a planning algorithm to compute a policy (for example, if the RL algorithm is model-based).

## 3 Defining sub-problems and hierarchical structure

We would like to capture the intuitive notion of modularity in the context of MDPs. Modularity allows a large problem to be broken down into sub-problems, which could be tackled and solved independently from the rest. Sub-problem solutions could then be "stitched" together to (approximately) solve the entire problem. Intuitively, if these sub-problems are relatively small and repeated, this approach can lead to large computational gains. We will now formalize these intuitions for MDPs.

**Definition 1** *Consider a partition of the non-terminal states $\mathcal{S} \setminus \{s_e\}$ into $L$ disjoint subsets $\mathcal{H} = \{\mathcal{S}_i\}_{i=1}^L$. We define an **induced** subMDP $\mathcal{M}_i = \langle \mathcal{S}_i \cup \mathcal{E}_i, \mathcal{A}, P_i, r_i, \mathcal{E}_i \rangle$ as follows:*

- $\mathcal{S}_i$ *is the **internal state set**, and the action space is still $\mathcal{A}$.*
- *The **exit state set** $\mathcal{E}_i$ is defined as $\mathcal{E}_i = \{e \in \mathcal{S} \setminus \mathcal{S}_i : \exists (s, a) \in \mathcal{S}_i \times \mathcal{A} \text{ s.t. } P(e|s, a) > 0\}$.*
- *The state space of $\mathcal{M}_i$ is $\mathcal{S}_i \cup \mathcal{E}_i$.*
- *$P_i$ and $r_i$ are respectively the restriction of $P$ and $r$ to domain $\mathcal{S}_i \times \mathcal{A}$.*
- *The subMDP $\mathcal{M}_i$ terminates once it reaches a state in $\mathcal{E}_i$ (i.e., an exit state).*

Given a partition $\mathcal{H}$ of $\mathcal{M}$, consider the set of induced subMDPs, $\{\mathcal{M}_i\}_{i=1}^L$. We define $M$, the maximum size of any subMDP, and $\mathcal{E}$, the set of all exit states, as follows:

$$M = \max_i |\mathcal{S}_i \cup \mathcal{E}_i| \quad \text{and} \quad \mathcal{E} = \cup_{i=1}^L \mathcal{E}_i, \tag{1}$$

where $|\cdot|$ denotes the cardinality of a set. Intuitively, each subMDP can be viewed as a sub-problem of the original MDP. We can exploit the fact that some sub-problems may be similar to each other, by solving only one instance and re-using this solution. We now define the notion of equivalent subMDPs, in order to capture this idea.

**Definition 2 (Equivalent subMDPs)** *Two subMDPs $\mathcal{M}_i$ and $\mathcal{M}_j$ are* equivalent *if there is a bijection $f : \mathcal{S}_i \cup \mathcal{E}_i \to \mathcal{S}_j \cup \mathcal{E}_j$ s.t. $f(\mathcal{S}_i) = \mathcal{S}_j$, $f(\mathcal{E}_i) = \mathcal{E}_j$, and, through $f$, the subMDPs have the same transition probabilities and rewards at internal states.*

Note that the constraints $f(\mathcal{S}_i) = \mathcal{S}_j$ and $f(\mathcal{E}_i) = \mathcal{E}_j$ ensure that an internal (or exit) state in $\mathcal{M}_i$ is mapped to an internal (or exit) state in $\mathcal{M}_j$. Let $K \leq L$ be the number of equivalence classes of subMDPs induced by a particular partition $\mathcal{H}$ of $\mathcal{M}$. When there is no repeatable structure, $K = L$. When the partition produces repeatable structure, $K < L$.

In summary, any state space partition $\mathcal{H}$ yields three parameters: the maximum size of an induced subMDP $M$, the number of subMDP equivalence classes $K$, and the total number of exit states $|\mathcal{E}|$. Our analyses will depend on these three quantities, rather than $|\mathcal{S}|$, the number of states in $\mathcal{M}$. While the results that we will present hold for *any* MDP $\mathcal{M}$ and *any* partition $\mathcal{H}$, they will offer dramatic improvements over the standard algorithms only if $\mathcal{M}$ **exhibits hierarchical structure with respect to a partition** $\mathcal{H}$, by which we mean:

1. $MK \ll |\mathcal{S}|$;
2. the number of exit states $|\mathcal{E}|$ is small relative to the total number of states $|\mathcal{S}|$.

Intuitively, condition 1 can be satisfied by having small $M$, small $K$ or both. If the number of equivalence classes $K$ is small, solving a subMDP once may produce solutions that can be re-used in many other parts of the original problem. If $M$ is small, all subMDPs have small size, so they would be relatively easy to solve. Finally, if $|\mathcal{E}|$ is small, intuitively we have a small number of states that connect the sub-problems. We can think of these as "bottleneck" states in $\mathcal{M}$, which have been shown before to enable computationally efficient planning (see e.g. Sutton et al. [1999], McGovern and Barto [2001], Stolle and Precup [2002], Simsek and Barto [2009], Solway et al. [2014]).

To make this notion of hierarchical structure more concrete, consider a garbage collecting robot navigating in a building. The robot's goal is to maximize the amount of trash it collects before exiting. The building has $L$ floors, and each floor belongs to one of $K$ floor "types" defined according to some criterion relevant to the robot. A floor of type $i$ has $|\mathcal{E}_i| \leq |\mathcal{E}|$ exits to other floors (elevators, stairs, etc.). When exiting a floor, the agent will either get to another floor or leave the building. Each floor has $L'$ rooms, which can also be grouped into $K'$ groups. Room type $j$ can be divided into $M'_j \leq M'$ regions that define the robot's current state; some of these regions contain garbage that should be collected. Rooms are connected through up to $|\mathcal{E}'|$ doors. If we think of the robot as an RL agent and the building as an MDP, the problem can be partitioned at two different levels: each subMDP can be either a floor or a room. Each of these partitions will result in different values for the constants appearing in items 1 and 2 above, which will in turn define how efficiently the agent can solve the problem.

In the next two sections, we will analyze RL (Sec. 4) and planning (Sec. 5), assuming that an agent starts with a known partition $\mathcal{H}$, in which the equivalence classes of the induced subMDPs are also known. We will establish results showing that leveraging hierarchical structure allows agents to achieve better statistical efficiency, in the case of learning, and better computational complexity, in the case of planning (at the cost of some controlled sub-optimality).

## 4 Statistically Efficient Learning with Hierarchical Structure

It is natural to think of hierarchical reinforcement learning as what an agent does when it possesses prior knowledge that the environment obeys hierarchical structure. As we will establish in this section, hierarchical structure can enable more efficient learning, and the difference can be dramatic if the MDP exhibits highly repetitive structure. This occurs when the number of subMDPs far exceeds the number of equivalence classes. We will formally characterize improvements in statistical efficiency through studying a specific reinforcement learning algorithm that can leverage prior knowledge in a coherent manner. We expect our qualitative insights to extend to other algorithms that carefully account for prior knowledge.

## 4.1 Posterior Sampling for Reinforcement Learning

Posterior sampling for reinforcement learning (PSRL), as introduced by Strens [2000] and analyzed in Osband et al. [2013] and Gopalan and Mannor [2015], offers an often effective approach to episodic reinforcement learning. Before each episode of interaction, the agent samples a model of the environment, possibly in the form of an MDP, from the posterior distribution over environments conditioned on data gathered over previous episodes. Then, the agent computes an optimal policy for the sampled model and applies that to select actions over the next episode. To guide exploration, PSRL relies on representation of epistemic uncertainty in terms of a probability distribution over environment. The prior distribution reflects the agent's initial partial knowledge about the environment. To quantify performance, regret bounds that apply under *any* prior distribution are established in Osband et al. [2013]. These bounds do not capture the benefits of greater degrees of prior knowledge, but subsequent results [Osband and Van Roy, 2014a,b] demonstrate that stronger regret bounds, reflecting dramatic improvements in agent performance, are possible when the prior distribution reflects knowledge of special environment structure.

Algorithm 1 offers pseudocode for a *generalized* form of PSRL. Over each episode, a sampler produces an MDP $\mathcal{M}^t$ that represents a statistically plausible model of the environment given the state of knowledge $\mathcal{P}^t$. Then, a planner computes a policy $\pi^t$ that approximately optimizes $\mathcal{M}^t$. This policy is executed over the episode, leading to a data set $\mathcal{D}_t$ made up of the trajectory of states, actions, and rewards. Finally, the state of knowledge is updated by an inference algorithm.

Ideally, as in the *pure* form of PSRL, $\mathcal{P}^t$ encodes a posterior distribution over MDPs, the sampling algorithm draws $\mathcal{M}^t$ from $\mathcal{P}^t$, the planner computes an optimal policy for $\mathcal{M}^t$, and the inference algorithm applies Bayes' rule. However, the more flexible generalization of Algorithm 1 can be applied more broadly, even when it is infeasible to exactly represent, compute, or sample from the posterior distribution or to identify an optimal policy.

---

**Algorithm 1:** PSRL with a Planner, Sampler, and Inferer

---

**Initialization:** prior knowledge $\mathcal{P}^0$, planning algorithm `plan`, sampling algorithm `sample`, inference algorithm `infer`;

**for** episode $t = 1, 2, \ldots T$ **do**
    sample $\mathcal{M}^t \sim \mathtt{sample}(\mathcal{P}^t)$;
    plan $\pi^t = \mathtt{plan}(\mathcal{M}^t)$;
    execute $\pi^t$ over episode $t$, observe $\mathcal{D}_t$;
    infer $\mathcal{P}^{t+1} = \mathtt{infer}(\mathcal{P}^t, \mathcal{D}_t)$ ;
**end**

---

## 4.2 Hierarchical Reinforcement Learning

We consider posterior sampling for hierarchical reinforcement learning (PSHRL) to be PSRL applied with a particular kind of prior distribution, and possibly with a planner that is customized for such an environment. In particular, we will consider priors that include only MDPs that obey hierarchical structure, as described earlier, for fixed values of $M$ and $K$. As for the planner, we will alternately consider an optimal planner and one designed to produce approximately optimal policies more efficiently by leveraging hierarchical structure, as we will discuss in Section 5.

The per-episode computational complexity of PSHRL depends on the special structure obeyed by $\mathcal{P}^t$ and $\mathcal{M}^t$, as well as the choice of `plan`, `sample`, and `infer`. As we will show in Section 5, suitable hierarchical structure can be leveraged to improve computational efficiency. The choice of $\mathcal{P}^0$, `sample`, and `infer` determine tractability of sampling and inference. Again, suitable hierarchical structure can allow for more efficient execution of these steps.

Recall that $K$ is the number of subMDP equivalence classes and $M$ is the maximal number of states per subMDP. Hierarchical structure is especially informative when $MK$ is small relative to the number of MDP states $|\mathcal{S}|$. This can yield dramatic improvements in statistical efficiency, as we will establish via a regret bound.

## 4.3 Regret Bound

For any learning algorithm `alg`, we define the Bayesian regret over the first $T$ episodes as

$$\text{BayesRegret}(\texttt{alg}, T) = \sum_{t=1}^{T} \mathbb{E}\left[ V^*(s_0) - V^{\pi^t}(s_0) \right], \qquad (2)$$

where $V^*$ is the optimal value function of $\mathcal{M}$, and $V^{\pi^t}$ is the value function under policy $\pi^t$. The regret is "Bayesian" in the sense that the expectation integrates over $\mathcal{M}$ with respect to the prior distribution $\mathcal{P}^0$. Note that minimizing $\text{BayesRegret}(\texttt{alg}, T)$ is equivalent to maximizing $\sum_{t=1}^{T} \mathbb{E}\left[ V^{\pi_t}(s_0) \right] = \sum_{t=1}^{T} \mathbb{E}\left[ \sum_{h=1}^{\tau_t} r_{th} \right]$. We will study $\text{BayesRegret}(\texttt{PSRL}, T)$, which is the Bayesian regret of `PSRL`, when applied with a prior that exhibits hierarchical structure.

Recall that, under any policy $\pi$, $\mathbb{E}[\tau] \leq H$ and $\tau \leq \tau_{\max}$ with probability 1. The following theorem is our main result on statistical efficiency.

**Theorem 1 (Regret Bound)** *If $\mathcal{P}^0$ exhibits hierarchical structure with a maximum of $M$ states per subMDP and $K$ subMDP equivalence classes,* `sample` *draws from the posterior distribution, and* `infer` *applies Bayes' rule, then*

$$\text{BayesRegret}(\texttt{PSRL}, T) \leq \underbrace{\mathbb{E}\left[ V^*(s_0) - V^{\tilde{\pi}}(s_0) \right] T}_{\textit{due to sub-optimal planning}} + \underbrace{O\left( H^{\frac{3}{2}} M \sqrt{K} \sqrt{|\mathcal{A}| T \log(|\mathcal{A}| K H \tau_{\max} T)} \right)}_{\textit{due to learning}},$$

*where $\tilde{\pi} = \texttt{plan}(\mathcal{M})$.*

Note that the expectation in $\mathbb{E}\left[ V^*(s_0) - V^{\tilde{\pi}}(s_0) \right]$ represents an integral with respect to the prior distribution $\mathcal{P}^0$ of $\mathcal{M}$. Also note that if `plan` exactly optimizes $\mathcal{M}$ then $\tilde{\pi} = \pi^*$, and therefore, $\mathbb{E}\left[ V^*(s_0) - V^{\tilde{\pi}}(s_0) \right] = 0$. Approximately optimal planning can contribute to regret a term $\mathbb{E}\left[ V^*(s_0) - V^{\tilde{\pi}}(s_0) \right] T$ that grows linearly with the number $T$ of episodes. One factor of $O(H)$ in the second term is due to the magnitude of optimal value $\mathbb{E}\left[ V^*(s_0) \right]$, which can grow with horizon, and is therefore inevitable.

We show that the hierarchical structure can enable statistically more efficient learning by comparing to the `PSRL` regret bound of Osband et al. [2013], which applies for any prior. Assuming that the MDP has fixed horizon $\tau = H$ and `plan` always returns an optimal policy, Osband et al. [2013] established $\text{BayesRegret}(\texttt{PSRL}, T) \leq \tilde{O}(H^{\frac{3}{2}} |\mathcal{S}| \sqrt{|\mathcal{A}| T})^2$, where the $\tilde{O}(\cdot)$ notation hides logarithmic factors. Our regret bound, under the same assumptions, is $\tilde{O}\left( H^{\frac{3}{2}} M \sqrt{K} \sqrt{|\mathcal{A}| T} \right)$, that is, we have replaced $\tilde{O}(|\mathcal{S}|)$ with $\tilde{O}(M \sqrt{K})$, which is highlighted in red font in Theorem 1. Notice that if $\mathcal{M}$ is treated as one subMDP, then we have $K = L = 1$ and $M = |\mathcal{S}|$, hence our regret bound reduces to that in Osband et al. [2013]. On the other hand, when $M \sqrt{K} \ll |\mathcal{S}|$, our regret bound conveys a dramatic improvement. This improvement can be interpreted in terms of two components. First, the replacement of $\tilde{O}(\sqrt{|\mathcal{S}|})$ with $\tilde{O}(\sqrt{MK})$ arises because the agent needs to learn about a smaller number of distinct states in the hierarchical MDP. Second, $\tilde{O}(\sqrt{|\mathcal{S}|})$ is replaced by $\tilde{O}(\sqrt{M})$ because at each state-action pair in the hierarchical MDP, the agent can transition to at most $M$ successor states.

The proof of Theorem 1 is partially motivated by analysis in Osband et al. [2013]. However, we consider a different setting and our results are technically more complex. Specifically, compared with Osband et al. [2013], Theorem 1 considers hierarchical structure, and allows for both sub-optimal planning and a random time horizon $\tau$. Please refer to Appendix A for the detailed proof of Theorem 1.

# 5  Computationally Efficient Planning with Hierarchical Structure

We now turn our attention to the problem of planning in $\mathcal{M}$ given a partition $\mathcal{H}$. This problem has been tackled in the framework of options, by using option models [Sutton et al., 1999]. Options can

**Algorithm 2:** Planning with Exit Profiles (PEP)

---

**Input:** MDP $\mathcal{M}$, $k$ sets of exit profiles $\tilde{\mathcal{J}}_k$, one for each equivalent subMDP class $k$;

**Step 1: Option generation**

**for** $k = 1, 2, \ldots K$ **do**

     For each exit profile $J \in \mathcal{J}_k$, compute one option $\pi_{k,J}$ for subMDPs in equivalence class $k$, and its associated model;

**end**

**Step 2: Plan with options**

Compute a policy for the induced high-level $\mathcal{M}^G$, which induces a policy $\tilde{\pi}$ for $\mathcal{M}$

**Return:** $\tilde{\pi}$

---

be viewed as policies which act in a subset of states, and with which one can associate temporally extended reward and transition models. The option policies can be thought of as solutions to sub-problems, generated by an MDP's structure, and some *subgoals*, which are additional rewards associated with particular states [Sutton et al., 1999]. One can view this approach as constructing options corresponding to subMDPs. To make this problem well defined, one needs to consider possible combinations of values associated with the exit states of a subMDP. For example, an agent which is in a room with two doors might consider making either door a subgoal, by giving it a high reward. An option can then be trained inside the room, which amasses treasure if it makes sense, then exits through the designated door. We now formalize this intuition through the notion of exit profiles.

**Definition 3 (Exit Profile)** *An* exit profile $J$ for subMDP $\mathcal{M}_i$ is a vector of values $J(e), \forall e \in \mathcal{E}_i$.

Note that from the perspective of a subMDP, the structure outside is summarized in an exit profile $J$. An exit profile induces an optimal policy for the subMDP $\mathcal{M}_i$, $\pi_{i,J}$, which we will think of as an *option*. By definition, an exit profile $J$ will induce the same option for equivalent subMDPs.

Once a set of options and associated models have been computed, one can define an induced **high-level MDP** $\mathcal{M}_G = \langle \mathcal{S}_G, \mathcal{A}_G, P^G, r^G \rangle$, whose state space $\mathcal{S}_G = \mathcal{E} \cup \{s_0\}$ is the union of all exit states and the initial state $s_0$. For each $s \in \mathcal{S}_G$, if $s$ is a state in a subMDP $\mathcal{M}_i$, then its action space $\mathcal{A}_G(s)$ is the set of options computed for $\mathcal{M}_i$. $r^G(s, \pi_{i,J})$ is the expected reward obtained from $s \in \mathcal{S}_i$ under option $\pi_{i,J}$ until this option reaches an exit state $e \in \mathcal{E}_i$, and $P^G(e|s, \pi_{i,J})$ gives the probability of transitioning to $e \in \mathcal{E}_i$. These quantities form the option model for $\pi_{i,J}$, defined as in Sutton et al. [1999], which can be computed at the same time as the option[3].

The process of creating a set of options and models corresponding to a set of exit profiles, then using them to solve $\mathcal{M}_G$, is summarized in Algorithm 2, which we call Planning with Exit Profiles (PEP).

## 5.1 Computational Complexity of Planning with Options

PEP is a blueprint for planning with options, which can be instantiated by using different dynamic programming approaches [Bertsekas, 2015] to implement steps 1 and 2. We will discuss the complexity of this algorithm when using value iteration (VI) for both steps, but similar analyses could be carried out easily for other algorithms (e.g., policy iteration).

In order to simplify the analysis and ensure that VI terminates in a finite number of steps, we make the following assumption:[4]

**Assumption 1** *For $\mathcal{M}$, all its induced subMDPs with exit profiles, and the induced $\mathcal{M}^G$, the transition probability graph corresponding to an optimal policy is acyclic.*

Under Assumption 1, VI will compute the value function in $n$ iterations under a proper initialization, where $n$ is the cardinality of the state space (see Section 3.4.1 of Bertsekas [2015]). Under this

assumption, the computational complexity of VI in $\mathcal{M}$ is $O(|\mathcal{S}|^2|\mathcal{A}|M)$, because by our definition, $M$ is an upper bound on the number of states into which any state-action pair can transition. Let $X = \max_k |\tilde{\mathcal{J}}_k|$ denote the maximum cardinality of an exit profile set over all equivalent subMDP classes (see Algorithm 2). The computational complexity of PEP can then be expressed as:

$$\underbrace{O(KXM^2|\mathcal{A}|M)}_{\text{for step 1}} + \underbrace{O(|\mathcal{E}|^2XM)}_{\text{for step 2}} \leq O\left(X[KM^2|\mathcal{A}| + |\mathcal{E}|^2]M\right). \tag{3}$$

Roughly speaking, planning with options will be efficient if $XM^2K < O(|\mathcal{S}|^2)$ and $|\mathcal{E}|^2X < O(|\mathcal{S}|^2|\mathcal{A}|)$, which means that all subMDPs are small, a small number of exit profiles are used to find options for each equivalent subMDP class, and the total number of exit states $|\mathcal{E}|$ is small.

## 5.2 Performance of Planning with Options

We now provide a performance bound for PEP, based on the "quality" of the exit profiles. Let $\mathcal{J}_i \subset [0, H]^{|\mathcal{E}_i|}$ be the space of possible exit profiles for subMDP $\mathcal{M}_i$. Let $V_{i,J}^\pi$ denote the value of a policy $\pi$ for an exit profile $J$ in $\mathcal{M}_i$. We denote $V_{i,J}^*$ the value of the optimal policy of $\mathcal{M}_i$ w.r.t. exit profile $J$, $\pi_{i,J}$. We now define the suboptimality of a set of exit profiles:

**Definition 4 (Exit Profile Suboptimality)** *The suboptimality of a set of exit profiles $\tilde{\mathcal{J}}$ for $\mathcal{M}_i$ is defined as:*

$$\Delta_i(\tilde{\mathcal{J}}) = \max_{s \in \mathcal{S}_i^0,\, J \in \mathcal{J}_i} \left[ V_{i,J}^*(s) - \max_{\tilde{J} \in \tilde{\mathcal{J}}} V_{i,J}^{\pi_{i,\tilde{J}}}(s) \right], \tag{4}$$

*where $S_i^0$ is the set of possible start states in $\mathcal{M}_i$, and $\mathcal{J}_i$ is the space of possible exit profiles.*

Notice that $V_{i,J}^{\pi_{i,\tilde{J}}}(s)$ is the value with exit profile $J$, under policy $\pi_{i,\tilde{J}}$ that is optimal for another exit profile $\tilde{J}$, at the start state $s$. In other words, the definition of $\Delta_i(\tilde{\mathcal{J}})$ ensures that for any exit profile $J$, there exists an exit profile in $\tilde{\mathcal{J}}$ that induces an $\Delta_i(\tilde{\mathcal{J}})$-optimal policy under $J$. Recall that in Algorithm 2, for subMDP $\mathcal{M}_i$ in equivalence class $k$, exit profiles $\tilde{\mathcal{J}}_k$ are used for option generation. Thus, we define $\Delta = \max_i \Delta_i(\tilde{\mathcal{J}}_{k_i})$, where $k_i$ is the equivalence class $\mathcal{M}_i$ is in. We can also prove that PEP with VI is near-optimal under Assumption 1 and a mild technical assumption.

**Proposition 1** *If in step 2 of PEP, the agent uses VI with initial $V = \mathbf{0}$ to compute a policy $\tilde{\pi}$, then under Assumption 1 and a mild technical assumption, we have: $V^*(s_0) - V^{\tilde{\pi}}(s_0) \leq \Delta|\mathcal{E}|$.*

Please refer to Appendix B.1 for the proof of this proposition. Roughly speaking, Proposition 1 states that if the total number of exit states, $|\mathcal{E}|$, is small, and the exit profiles used in PEP have high quality ($\Delta$ is small), then PEP returns a near-optimal policy.

## 5.3 Sufficient Conditions for High-Quality Exit Profiles

The previous results indicate that in order to ensure that planning with options is both computationally efficient and returns a near-optimal policy, we need the set of exit profiles considered by PEP to have small cardinality (for computational efficiency) and high quality (for near-optimality). We now discuss how to choose such exit profile sets.

In general, an exit profile set can be chosen based on an $\epsilon$-**cover**, defined as follows: a finite set $\tilde{\mathcal{J}}$ is an $\epsilon$-cover for $\mathcal{J}$ if for any $J \in \mathcal{J}$, there exists $\tilde{J} \in \tilde{\mathcal{J}}$ s.t. $\|J - \tilde{J}\|_\infty \leq \epsilon$. We then have the following result:

**Proposition 2** *For subMDP $\mathcal{M}_i$ in equivalence class $k$, if $\tilde{\mathcal{J}}_k$ is an $\epsilon$-cover for $\mathcal{J}_i$, then $\Delta_i(\tilde{\mathcal{J}}_k) \leq 2\epsilon$.*

Please refer to Appendix B.2 for the proof of Proposition 2. Since $\mathcal{J}_i \subseteq [0, H]^{|\mathcal{E}_i|}$, there always exists a finite $\epsilon$-cover $\tilde{\mathcal{J}}_k$ for $\mathcal{J}_i$ with $|\tilde{\mathcal{J}}_k| \leq \lceil \frac{H}{\epsilon} \rceil^{|\mathcal{E}_i|}$. In general, the cardinality of an $\epsilon$-cover is too large to guarantee PEP's computational efficiency. However, if $\max_i |\mathcal{E}_i|$ is very small (e.g. $\max_i |\mathcal{E}_i| \leq 3$), then PEP with $\epsilon$-cover will be computationally efficient.

Another favorable special case is when subMDP $\mathcal{M}_i$ has **deterministic exiting**. That is, with any start state $s$ and under any deterministic policy $\pi$, the agent exits $\mathcal{M}_i$ at a single state $e_{s,\pi} \in \mathcal{E}_i$ with

probability 1. Note that in general $e_{s,\pi}$ depends on both the start state $s$ and the deterministic policy $\pi$. Deterministic exiting will occur if $\mathcal{M}_i$ has only one exit state, or deterministic transitions. For any $e \in \mathcal{E}_i$, we define $J_e : \mathcal{E}_i \to \Re$ as $J_e(s) = (H+1)\mathbf{1}[s=e]$, where $\mathbf{1}[\cdot]$ is the indicator function. We then have the following result:

**Proposition 3** *For subMDP $\mathcal{M}_i$ in equivalence class $k$, if $\mathcal{M}_i$ has deterministic exiting and $\tilde{\mathcal{J}}_k = \{J_e : e \in \mathcal{E}_i\}$, then $\Delta_i(\tilde{\mathcal{J}}_k) = 0$.*

Please refer to Appendix B.3 for the proof of Proposition 3. Note that in this case $|\tilde{\mathcal{J}}_k| = |\mathcal{E}_i|$. Based on Proposition 1, if all the subMDPs have deterministic exiting, then we can efficiently compute an optimal policy $\pi^*$ for $\mathcal{M}$ by using PEP.

## 6 Summary of Results

| learning alg. | planner | regret bound | computation per episode |
|---|---|---|---|
| PSRL | VI | $\tilde{O}(H^{\frac{3}{2}}\lvert\mathcal{S}\rvert\sqrt{\lvert\mathcal{A}\rvert T})$ | $O(\lvert\mathcal{S}\rvert^2\lvert\mathcal{A}\rvert M)$ |
| PSHRL | VI | $\tilde{O}(H^{\frac{3}{2}}M\sqrt{K}\sqrt{\lvert\mathcal{A}\rvert T})$ | $O(\lvert\mathcal{S}\rvert^2\lvert\mathcal{A}\rvert M)$ |
| PSHRL | PEP | $\Delta\lvert\mathcal{E}\rvert T + \tilde{O}(H^{\frac{3}{2}}M\sqrt{K}\sqrt{\lvert\mathcal{A}\rvert T})$ | $O(X(M^2K\lvert\mathcal{A}\rvert + \lvert\mathcal{E}\rvert^2)M)$ |

Table 1: Algorithm Comparison. Differences in regret bounds and computational complexities are highlighted in red font. Recall that $\mathcal{S}$ and $\mathcal{A}$ are the state and action space of $\mathcal{M}$; $H$ is a bound on the expected time horizon of $\mathcal{M}$; $T$ is the number of interaction episodes; $M$ is the maximum subMDP size; $K$ is the number of subMDP equivalence classes; $\mathcal{E}$ is the set of all exit states; $\Delta$ and $X$ respectively measure the quality and the number of exit profiles used in PEP.

In Table 1, we summarize the regret bounds and per-episode computational complexities of three algorithms: (1) PSRL with value iteration (VI) planning, (2) PSHRL with VI planning, and (3) PSHRL with PEP planning[5]. As discussed before, if a partition with many repeated, small subMDPs is available, the regret bound for (2) is much smaller than (1), since PSHRL exploits hierarchical structure during learning. Moreover, if all the subMDPs also have few exit states and admit a small set of high-quality exit profiles, then (3) will be computationally much more efficient than (1) and (2), and only incurs an additional $O(\Delta\lvert\mathcal{E}\rvert T)$ regret compared with (2), due to sub-optimal planning.

## 7 Related work

Several works have tackled decompositions of MDPs into sub-problems in a classical planning context [Dean and Lin, 1995, Singh and Cohn, 1998, Meuleau et al., 1998]. The closest related to our work is the framework of weakly coupled MDPs [Meuleau et al., 1998], which considers an MDP decomposition into subMDPs, which are then solved independently by planning and whose solutions are related through a set of constraints. Loose coupling of the sub-problems allows a small set of such constraints, which intuitively has the same effect as our exit profiles. Sub-tasks with a small number of exits have been modelled through the notion of bottleneck states [McGovern and Barto, 2001, Stolle and Precup, 2002, Simsek and Barto, 2009, Solway et al., 2014]. According to our analysis, the existence of such states would indeed imply a small number of exit profiles, and hence efficient planning. Some existing work, e.g. Solway et al. [2014], Harutyunyan et al. [2019], has proposed optimization objectives for identifying such states using information-related criteria.

It is worth pointing out that the "equivalent subMDP" notion used in this paper can be further relaxed. In particular, rather than assuming that subMDPs in the same class have the *same* reward/transition models, we can assume that they have *similar* reward/transition models. Most results in this paper can be extended to that case, by leveraging results from planning with approximate models [Jiang et al., 2016], such as approximate MDP homomorphisms [Ravindran and Barto, 2004], or using bisimulation metrics to assess the similarity of subMDPs within a class [Ferns et al., 2004]. We leave

such extensions for future work. Our notion of subMDPs and exit profiles can also be used to model the use of HRL for transfer learning (see [Taylor and Stone, 2009] for an overview of the latter topic, which includes both HRL as well as methods for state equivalence).

Finally, we briefly compare this paper with Mann et al. [2015]. To summarize, Mann et al. [2015] discuss and analyze two algorithms: *Option-Fitted Value Iteration (OFVI)* and *Landmark-Approximate Value iteration (LAVI)*. The OFVI analysis relies on the discounted-average concentrability of the future state distributions in the semi-MDP defined by options, so it is a very different-flavor result. LAVI relies on options that go to designated landmark states, and which are computed by solving a deterministic relaxation of the semi-MDP in a neighborhood of landmarks. In our terminology, such options have a single exit state, and LAVI then solves the problem that jumps between landmarks. There is no repeating structure in this approach; in fact, each option only applies in a small neighborhood of state space around a landmark. Our result could be applied to the LAVI setup directly, but it would be hard to compare to their bound directly due to the very different quantities involved.

## 8 Concluding Remarks

We have presented two theoretical results which illuminate a kind of problem structure that benefits HRL algorithms. Briefly, the ability to partition an MDP into repeating subMDPs leads to improved regret bounds for a Thompson sampling-style algorithm that takes advantage of this structure. Furthermore, we highlighted and formalized a trade-off between the quality and complexity of planning in MDPs where the structure allows for a small number of problems to be solved. The insights provided in this work can be used not just to further theoretical understanding of HRL, but also for two immediate practical goals. First, one could use the structure of MDP we introduced as a template to build examples to study HRL algorithms under controlled conditions [Osband et al., 2020]. Second, these results can be turned into objective functions for HRL algorithms that *discover* the right structure. This could lead, for example, to algorithms for the discovery of options endowed with initiation sets (corresponding to the subMDP partitions), or to anytime planning algorithms that construct exit profiles as needed. Ultimately, we would like to put together the learning and planning results into a theoretical framework that elucidates the trade-off of three components that are crucial to RL agents: solution quality (as expressed by the true value of the policy found), sample complexity, and computational complexity. This trade-off is enabled by HRL, and can be understood through the lens of the analysis we presented: HRL allows controlling the complexity of the policy class of an agent, and this control should take into account both the agent's ability to acquire data and its available computational resources. We hope to develop this line of inquiry in future work.

## Broader Impact

This is a theoretical investigation and as such does not present any foreseeable immediate societal impact beyond the general concerns over progress in artificial intelligence. Specifically, we focused on the question of how to design efficient hierarchical agents for reinforcement learning problems with repeating sub-problem structure. The insights provided have the potential to help practitioners to build more efficient hierarchical agents for real-world reinforcement learning problems, whose societal impact depends on the specific application.

## Acknowledgments and Disclosure of Funding

All authors of this paper are employees of DeepMind. This paper is not supported by any external funding.

## Footnotes

[1] Note that the fixed initial state assumption can be easily relaxed to allow for an initial state drawn from any fixed distribution.

[2]Some notations in Osband et al. [2013] have different meanings. Specifically, "$\tau$" in Osband et al. [2013] means $H$ and "$T$" in Osband et al. [2013] means $HT$.

[3]Note that we have no discount factor, due to the finite-time horizon, which simplifies the model and gives rise to an MDP at the high level as well, instead of an SMDP.

[4]We make Assumption 1 to simplify the exposition of the computational complexity results. This assumption is not strictly necessary and can be relaxed.

[5]We assume that the chosen `sample` and `infer` in Algorithm 1 can be executed efficiently and most computation is due to `plan`, as is typical in model-based RL.

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
