[Supplementary Material]

# Appendices

## A  Proof for Theorem 1

Before proceeding, let us define an additional term $\bar{S} = \sum_{k=1}^{K} |\mathcal{S}_{i_k}|$, which is the sum over equivalence classes of the number of internal states in a subMDP from each class. Intuitively, $\bar{S}$ can be thought of as the number of distinct internal states. Note that trivially, we have $\bar{S} \leq KM$.

For any MDP $\tilde{\mathcal{M}}$ consistent with the prior $\mathcal{P}^0$ and any policy $\pi : \mathcal{S} \to \mathcal{A}$, we use $V^{\tilde{\mathcal{M}},\pi}$ to denote the expected total reward in $\tilde{\mathcal{M}}$ under policy $\pi$, with initial state $s_0$. Then by definition, we have

$$
\begin{aligned}
\mathrm{BayesRegret}(\mathtt{PSHRL}, T) &= \sum_{t=1}^{T} \mathbb{E}\left[ V^{\mathcal{M},\pi^*} - V^{\mathcal{M},\pi^t} \right] \\
&= \sum_{t=1}^{T} \mathbb{E}\left[ V^{\mathcal{M},\pi^*} - V^{\mathcal{M},\tilde{\pi}} \right] + \sum_{t=1}^{T} \mathbb{E}\left[ V^{\mathcal{M},\tilde{\pi}} - V^{\mathcal{M},\pi^t} \right] \\
&= \mathbb{E}\left[ V^{\mathcal{M},\pi^*} - V^{\mathcal{M},\tilde{\pi}} \right] T + \sum_{t=1}^{T} \mathbb{E}\left[ V^{\mathcal{M},\tilde{\pi}} - V^{\mathcal{M},\pi^t} \right], \quad (5)
\end{aligned}
$$

where the last equality follows from the fact that $V^{\mathcal{M},\pi^*}$ and $V^{\mathcal{M},\tilde{\pi}}$ do not depend on $t$. Let $\mathcal{H}_t$ denote the "history" at the start of episode $t$, which includes all the observations by the start of episode $t$. Notice that conditioning on $\mathcal{H}_t$, $\mathcal{M}_t$ and $\mathcal{M}$ are i.i.d. Since by definition, $\tilde{\pi} = \mathtt{plan}(\mathcal{M})$, $\pi^t = \mathtt{plan}(\mathcal{M}^t)$, and $\mathtt{plan}$ is a deterministic mapping, thus, conditioning on $\mathcal{H}_t$, $(\mathcal{M}, \tilde{\pi})$ and $(\mathcal{M}^t, \pi^t)$ are also i.i.d. So we have $\mathbb{E}\left[ V^{\mathcal{M},\tilde{\pi}} \middle| \mathcal{H}_t \right] = \mathbb{E}\left[ V^{\mathcal{M}^t,\pi^t} \middle| \mathcal{H}_t \right]$, which implies that

$$
\mathbb{E}\left[ V^{\mathcal{M},\tilde{\pi}} \right] = \mathbb{E}\left[ \mathbb{E}\left[ V^{\mathcal{M},\tilde{\pi}} \middle| \mathcal{H}_t \right] \right] = \mathbb{E}\left[ \mathbb{E}\left[ V^{\mathcal{M}^t,\pi^t} \middle| \mathcal{H}_t \right] \right] = \mathbb{E}\left[ V^{\mathcal{M}^t,\pi^t} \right].
$$

Thus, we have

$$
\sum_{t=1}^{T} \mathbb{E}\left[ V^{\mathcal{M},\tilde{\pi}} - V^{\mathcal{M},\pi^t} \right] = \sum_{t=1}^{T} \mathbb{E}\left[ V^{\mathcal{M}^t,\pi^t} - V^{\mathcal{M},\pi^t} \right].
$$

For any policy $\pi$ and any MDP $\tilde{\mathcal{M}}$, we use $\mathcal{T}^{\tilde{M},\pi}$ to denote the dynamic programming operator in $\tilde{\mathcal{M}}$ under $\pi$. In other words, the Bellman equation in any MDP $\tilde{\mathcal{M}}$ under any policy $\pi$ is $V^{\tilde{\mathcal{M}},\pi} = \mathcal{T}^{\tilde{\mathcal{M}},\pi} V^{\tilde{\mathcal{M}},\pi}$. Then from Section 5.1 of Osband et al. [2013], we have

$$
\mathbb{E}\left[ V^{\mathcal{M}^t,\pi^t} - V^{\mathcal{M},\pi^t} \middle| \mathcal{M}^t, \mathcal{M} \right] = \mathbb{E}\left[ \sum_{h=1}^{\tau_t-1} \left( \mathcal{T}^{\mathcal{M}^t,\pi^t} - \mathcal{T}^{\mathcal{M},\pi^t} \right) V^{\mathcal{M}^t,\pi_t}(s_{th}) \middle| \mathcal{M}, \mathcal{M}^t \right],
$$

where $s_{th}$'s are generated under policy $\pi^t$ in the real MDP $\mathcal{M}$. The above equation decomposes the per-episode regret into one-step Bellman errors.

We now construct a high-probability confidence set. For any two non-terminal states $s, s' \in \mathcal{S}$, we say $s$ and $s'$ are **equivalent states** if they are internal states in two equivalent subMDPs, and the bijection between these two subMDPs maps $s$ to $s'$. Obviously, the notion of equivalent states is transitive. Let $\{\mathcal{X}_k\}_k$ be a partition of $\mathcal{S}$ based on equivalent states. That is, each $\mathcal{X}_k$ is an equivalent state class. By definition, we have $|\{\mathcal{X}_k\}_k| = \bar{S}$. For each episode $t$, let $N^t(k, a)$ denote the number of times action $a$ has been chosen at a state in the **equivalent state class** $k$ in the first $t-1$ episodes. We also use $\hat{P}^t$ and $\hat{r}^t$ to respectively denote the empirical transition model and the empirical average reward based on observations in the first $t-1$ episodes. Specifically,

- $\hat{P}^t(\cdot|s,a)$ and $\hat{r}^t(s,a)$ are estimated based on observations of choosing action $a$ at state $s$ **or its equivalent states**.

- If $N^t(s,a) = 0$, $\hat{P}^t(\cdot|s,a)$ and $\hat{r}^t(s,a)$ are not well defined. In this case, we choose $\hat{r}^t(s,a)$ as an arbitrary number in $[0,1]$ and $\hat{P}^t(\cdot|s,a)$ as an arbitrary distribution subject to the constraint that $\hat{P}^t(s'|s,a) > 0$ only if $s'$ and $s$ are in the same subMDP.

Recall that the prior $\mathcal{P}^0$, and hence all the posteriors $\mathcal{P}^t$, encodes the hierarchical information about equivalent subMDPs. Consequently, the PSHRL algorithm will only sample MDPs satisfying this equivalent subMDP restriction. Thus, we choose the confidence set at episode $t$ as:

$$\mathbb{M}_t = \left\{ \tilde{\mathcal{M}} : \left\| \hat{P}^t(\cdot|s,a) - P_k^{\tilde{\mathcal{M}}}(\cdot|s,a) \right\|_1 \leq \beta_1\left(N^t(k_s,a),t\right) \; \forall s,a, \right.$$

$$\left| \hat{r}^t(s,a) - \bar{r}^{\tilde{\mathcal{M}}}(s,a) \right| \leq \beta_2\left(N^t(k_s,a),t\right) \quad \forall s,a,$$

$$\left. \text{and } \tilde{\mathcal{M}} \text{ satisfies the equivalent subMDP restriction} \right\}, \tag{6}$$

where $k_s$ is the equivalent state class that state $s$ is in. Let $A = |\mathcal{A}|$, and recall that $M = \max_i |\mathcal{S}_i \cup \mathcal{E}_i|$, we have the following lemma:

**Lemma 1** *For any $\delta \in (0,1)$, if we choose $\beta_1$ and $\beta_2$ as $\beta_1(n,t) = \sqrt{\frac{14M \log\left(\frac{2AK\tau_{\max}t}{\delta}\right)}{\max\{1,n\}}}$ and $\beta_2(n,t) = \sqrt{\frac{7 \log\left(\frac{2MAK\tau_{\max}t}{\delta}\right)}{2\max\{1,n\}}}$, then we have*

$$P(\mathcal{M} \notin \mathbb{M}_t) = P(\mathcal{M}^t \notin \mathbb{M}_t) \leq \frac{\delta}{15t^6}.$$

**Proof:** This lemma is based on Lemma 17 of Jaksch et al. [2010], which is based on the following two results:

- $L_1$-**deviation of the true distribution and the empirical distribution:** Assume $p(\cdot)$ is a distribution over $m$ distinct events and $\hat{p}(\cdot)$ is an empirical distribution for $p$ from $n$ i.i.d. samples. From Theorem 2.1 in Weissman et al. [2003], for any $\epsilon > 0$, we have

$$P\left\{ \|p(\cdot) - \hat{p}(\cdot)\|_1 \geq \epsilon \right\} \leq (2^m - 2) \exp\left(-\frac{n\epsilon^2}{2}\right). \tag{7}$$

- **Hoeffding's inequality:** For the deviation between the true mean $\bar{r}$ and the empirical mean $\hat{r}$ from $n$ i.i.d. samples with support in $[0,1]$, for any $\epsilon \geq 0$, we have

$$P\left\{ |\bar{r} - \hat{r}| \geq \epsilon \right\} \leq 2\exp(-2n\epsilon^2).$$

Notice that at any state $s$, under action $a$, based on the definition of $M$, the agent might transit to at most $M$ states. Thus, in this case we can use inequality 7 with $m = M$. Assume that $\hat{P}^t(\cdot|s,a)$ is an empirical distribution based on $n \geq 1$ i.i.d. samples from the true distribution $P^{\mathcal{M}}(\cdot|s,a)$, then from Lemma 17 of Jaksch et al. [2010], by choosing $\beta_1(n,t) = \sqrt{\frac{14M \log\left(\frac{2AK\tau_{\max}t}{\delta}\right)}{\max\{1,n\}}}$, we have

$$P\left( \left\| \hat{P}^t(\cdot|s,a) - P^{\mathcal{M}}(\cdot|s,a) \right\|_1 \geq \beta_1(n,t) \middle| \mathcal{M}, n \text{ i.i.d. samples} \right) \leq \frac{\delta}{20t^7 M A \tau_{\max} K}.$$

On the other hand, based on the Hoeffding's inequality, if we choose $\beta_2(n,t) = \sqrt{\frac{7 \log\left(\frac{2MAK\tau_{\max}t}{\delta}\right)}{2\max\{1,n\}}}$, we have

$$P\left( \left| \hat{r}_k^t(s,a,h) - r_k^{\mathcal{M}}(s,a,h) \right| \geq \beta_2(n,t) \middle| \mathcal{M}, n \text{ i.i.d. samples} \right) \leq \frac{\delta}{60t^7 M A \tau_{\max} K}.$$

Notice that in episode $t$, $N^t(s,a)$ is a random variable that can take values $0, 1, \ldots, (t-1)(\tau_{\max}-1)$ (recall that each episode has horizon $\tau \leq \tau_{\max}$ with probability 1, and the last state is always $s_e$). Based on our definitions of $\beta_1$ and $\beta_2$, for $n = 0$ (the case without observations), the confidence intervals trivially hold with probability 1. Thus, union bound over possible values of $N^t(k_s,a)$ gives

$$P\left( \left\| \hat{P}^t(\cdot|s,a) - P^{\mathcal{M}}(\cdot|s,a) \right\|_1 \geq \beta_1(N^t(k_s,a),t) \middle| \mathcal{M} \right) \leq \sum_{n=1}^{t\tau_{\max}} \frac{\delta}{20t^7 M A \tau_{\max} K} < \frac{\delta}{20t^6 M A K}$$

$$P\left( \left| \hat{r}^t(s,a) - r^{\mathcal{M}}(s,a) \right| \geq \beta_2(N^t(k_s,a),t) \middle| \mathcal{M} \right) \leq \sum_{n=1}^{t\tau_{\max}} \frac{\delta}{60t^7 M A \tau_{\max} K} < \frac{\delta}{60t^6 M A K}$$

Notice that there are $A$ actions and at most $MK$ equivalent state classes. Taking a union bound over actions and equivalent state classes, we have

$$P(\mathcal{M} \notin \mathbb{M}_t | \mathcal{M}) < MAK \left[ \frac{\delta}{60t^6 MAK} + \frac{\delta}{20t^6 MAK} \right] = \frac{\delta}{15t^6}.$$

Since the above result holds for any $\mathcal{M}$, we have

$$P(\mathcal{M} \notin \mathbb{M}_t) = \sum_{\mathcal{M}} P(\mathcal{M}) P(\mathcal{M} \notin \mathbb{M}_t | \mathcal{M}) < \frac{\delta}{15t^6}.$$

Since $\mathcal{M}^t$ and $\mathcal{M}$ are conditionally i.i.d. given $\mathcal{H}_t$, we have

$$P(\mathcal{M}^t \notin \mathbb{M}_t) = \sum_{\mathcal{H}_t} P(\mathcal{H}_t) P(\mathcal{M}^t \notin \mathbb{M}_t | \mathcal{H}_t) = \sum_{\mathcal{H}_t} P(\mathcal{H}_t) P(\mathcal{M} \notin \mathbb{M}_t | \mathcal{H}_t) = P(\mathcal{M} \notin \mathbb{M}_t).$$

This concludes the proof. **q.e.d.**

Note that for any $\tilde{M}$ that can be sampled from the prior and any policy $\pi$, we have naive bounds on $V^{\tilde{\mathcal{M}},\pi}(s)$. To see it, recall that we assume $\mathbb{E}[\tau] \le H$ for any initial state $s \in \mathcal{S}$ and the reward support is a subset of $[0,1]$, thus we have $0 \le V^{\tilde{\mathcal{M}},\pi}(s) \le H$ for all $s \in \mathcal{S}$. Thus, we have:

$$\sum_{t=1}^{T} \mathbb{E}\left[ V^{\mathcal{M}^t,\pi^t} - V^{\mathcal{M},\pi^t} \right] \le \sum_{t=1}^{T} \mathbb{E}\left[ \left( V^{\mathcal{M}^t,\pi^t} - V^{\mathcal{M},\pi^t} \right) \mathbf{1}\left[ \mathcal{M}, \mathcal{M}^t \in \mathbb{M}_t \right] \right]$$

$$+ 2H \sum_{t=1}^{T} P(\mathcal{M} \notin \mathbb{M}_t), \tag{8}$$

Notice that by choosing $\delta = \frac{1}{H}$, we have

$$2H \sum_{t=1}^{T} P(\mathcal{M} \notin \mathbb{M}_t) < 2H \sum_{t=1}^{T} \frac{1}{15Ht^6} = \frac{2}{15} \sum_{t=1}^{T} \frac{1}{t^6} \le \frac{2}{15} \sum_{t=1}^{\infty} \frac{1}{t^2} < \frac{1}{3}.$$

On the other hand, we have

$$\sum_{t=1}^{T} \mathbb{E}\left[ \left( V^{\mathcal{M}^t,\pi^t} - V^{\mathcal{M},\pi^t} \right) \mathbf{1}\left[ \mathcal{M}, \mathcal{M}^t \in \mathbb{M}_t \right] \right]$$

$$= \sum_{t=1}^{T} \left\{ \mathbb{E}\left[ \sum_{h=1}^{\tau_t - 1} \left( \mathcal{T}^{\mathcal{M}^t,\pi^t} - \mathcal{T}^{\mathcal{M},\pi^t} \right) V^{\mathcal{M}^t,\pi_t}(s_{th}) \middle| \mathcal{M}, \mathcal{M}^t \right] \mathbf{1}\left[ \mathcal{M}, \mathcal{M}^t \in \mathbb{M}_t \right] \right\} \tag{9}$$

Notice that if $\mathcal{M}, \mathcal{M}^t \in \mathbb{M}_t$, we have

$$\left| \left( \mathcal{T}^{\mathcal{M}^t,\pi^t} - \mathcal{T}^{\mathcal{M},\pi^t} \right) V^{\mathcal{M}^t,\pi_t}(s_{th}) \right| \le \left| \bar{r}^{\mathcal{M}^t}\left( s_{th}, \pi^t(s_{th}) \right) - \bar{r}^{\mathcal{M}}\left( s_{th}, \pi^t(s_{th}) \right) \right|$$

$$+ \left\| P^{\mathcal{M}_t}(\cdot | s_{th}, \pi^t(s_{th})) - P^{\mathcal{M}}(\cdot | s_{th}, \pi^t(s_{th})) \right\|_1 \cdot \left\| V^{\mathcal{M}^t,\pi_t} \right\|_{\infty}$$

$$\le 2\beta_2(N^t(k_{s_{th}}, a_{th}), t) + 2\beta_1(N^t k_{s_{th}}, a_{th}), t)H \tag{10}$$

To simplify the exposition, we use $k_{th}$ to denote $k_{s_{th}}$. Hence, we have

$$\sum_{t=1}^{T} \mathbb{E}\left[ \left( V^{\mathcal{M}^t,\pi^t} - V^{\mathcal{M},\pi^t} \right) \mathbf{1}\left[ \mathcal{M}, \mathcal{M}^t \in \mathbb{M}_t \right] \right]$$

$$\le 2 \sum_{t=1}^{T} \mathbb{E}\left\{ \sum_{h=1}^{\tau_t - 1} \left[ \beta_2(N^t(k_{tk}, a_{tk}), t) + \beta_1(N^t(k_{tk}, a_{tk}), t)H \right] \right\}.$$

Notice that $t \le T$ always holds, with $\delta = \frac{1}{H}$, we have

$$\beta_2(N^t(k_{th}, a_{th}), t) + \beta_1(N^t(k_{th}, a_{th}), t)H \le O\left( H\sqrt{\frac{M \log(AKH\tau_{\max}T)}{\max\{1, N^t(k_{th}, a_{th})\}}} \right).$$

Finally, we provide "self-normalization" bounds for

$$\mathbb{E}\left\{\sum_{t=1}^{T}\sum_{h=1}^{\tau_t-1}\sqrt{\frac{1}{\max\left\{1, N^t(k_{th}, a_{th})\right\}}}\right\}.$$

Notice that

$$\sum_{t=1}^{T}\sum_{h=1}^{\tau_t-1}\sqrt{\frac{1}{\max\left\{1, N^t(k_{th}, a_{th})\right\}}} = \sum_{(k,a)}\sum_{t=1}^{T}\sum_{h=1}^{\tau_t-1}\sqrt{\frac{\mathbf{1}[(k_{th}, a_{th}) = (k, a)]}{\max\left\{1, N^t(k, a)\right\}}}$$

For any $(k, a)$, we have

$$\sum_{t=1}^{T}\sum_{h}^{\tau_t-1}\sqrt{\frac{\mathbf{1}[(k_{th}, a_{th}) = (k, a)]}{\max\left\{1, N^t(k, a)\right\}}} = \sum_{t=1}^{T}\sum_{h}^{\tau_t-1}\sqrt{\frac{\mathbf{1}[(k_{th}, a_{th}) = (k, a)]}{\max\left\{1, N^t(k, a)\right\}}}\mathbf{1}\left[N^t(k, a) \le \tau_{\max}\right]$$

$$+ \sum_{t=1}^{T}\sum_{h}^{\tau_t-1}\sqrt{\frac{\mathbf{1}[(k_{th}, a_{th}) = (k, a)]}{\max\left\{1, N^t(k, a)\right\}}}\mathbf{1}\left[N^t(k, a) > \tau_{\max}\right]$$

$$\overset{(a)}{<} 2\tau_{\max} + \sum_{n=1}^{N^{T+1}(k,a)}\frac{\sqrt{2}}{\sqrt{n}}$$

$$< 2\tau_{\max} + \int_{0}^{N^{T+1}(k,a)}\frac{\sqrt{2}}{\sqrt{n}}dn = 2\tau_{\max} + 2\sqrt{2N^{T+1}(k, a)},$$

$$(11)$$

where inequality (a) follows from the following observations:

- Since in each episode has maximum horizon $\tau_{\max}$, and $N_t(k, a)$ will be updated at the end of each episode $t$, then we have

$$\sum_{t=1}^{T}\sum_{h}^{\tau_t-1}\sqrt{\frac{\mathbf{1}[(k_{th}, a_{th}) = (k, a)]}{\max\left\{1, N^t(k, a)\right\}}}\mathbf{1}\left[N^t(k, a) \le \tau_{\max}\right]$$

$$\le \sum_{t=1}^{T}\sum_{h}^{\tau_t-1}\mathbf{1}[(k_{th}, a_{th}) = (k, a)]\mathbf{1}\left[N^t(k, a) \le \tau_{\max}\right] \le 2\tau_{\max}. \qquad (12)$$

- Assume $N^t(k, a) > \tau_{\max}$, and assume that $(k, a)$ has been interacted for $j_t \le \tau_t$ times in episode $t$, then, in episode $t$ we have

$$\sum_{h}^{\tau_t-1}\sqrt{\frac{\mathbf{1}[(k_{th}, a_{th}) = (k, a)]}{\max\left\{1, N^t(k, a)\right\}}}\mathbf{1}\left[N^t(k, a) > \tau_{\max}\right] \le \sum_{j=1}^{j_t}\sqrt{\frac{2}{N^t(k, a) + j}}\mathbf{1}\left[N^t(k, a) > \tau_{\max}\right],$$

which follows from the inequality $\frac{1}{n} \le \frac{2}{n+j}$ for $n > \tau_{\max} \ge \tau_t$ and $j < \tau_t$. Hence, we have

$$\sum_{t=1}^{T}\sum_{h}^{\tau_t-1}\sqrt{\frac{\mathbf{1}[(k_{th}, a_{th}) = (k, a)]}{\max\left\{1, N^t(k, a)\right\}}}\mathbf{1}\left[N^t(k, a) > \tau_{\max}\right] \le \sum_{n=1}^{N^{T+1}(k,a)}\sqrt{\frac{2}{n}}.$$

Thus we have

$$\sum_{(k,a)}\sum_{t=1}^{T}\sum_{h=1}^{T}\sqrt{\frac{\mathbf{1}[(k_{th}, a_{th}) = (k, a)]}{\max\left\{1, N^t(k, a)\right\}}} \le 2\tau_{\max}\bar{S}A + 2\sqrt{2}\sum_{k,a}\sqrt{N^{T+1}(k, a)}$$

$$\overset{(b)}{\le} 2\tau_{\max}\bar{S}A + 2\sqrt{2}\sqrt{\bar{S}A}\sqrt{\sum_{(k,a)}N^{T+1}(k, a)}$$

where (b) follows from the Cauchy-Schwarz inequality. Hence, we have

$$\mathbb{E}\left\{\sum_{t=1}^{T}\sum_{h=1}^{\tau_t-1}\sqrt{\frac{1}{\max\left\{1,N^t(k_{th},a_{th})\right\}}}\right\} \le 2\tau_{\max}\bar{S}A + 2\sqrt{2}\sqrt{\bar{S}A}\mathbb{E}\left[\sqrt{\sum_{(k,a)}N^{T+1}(k,a)}\right]$$

$$\le 2\tau_{\max}\bar{S}A + 2\sqrt{2}\sqrt{\bar{S}A}\sqrt{\mathbb{E}\left[\sum_{(k,a)}N^{T+1}(k,a)\right]}$$

$$\le 2\tau_{\max}\bar{S}A + 2\sqrt{2}\sqrt{\bar{S}A}\sqrt{\sum_{t=1}^{T}\mathbb{E}\left[\tau_t\right]}$$

$$\le 2\tau_{\max}\bar{S}A + 2\sqrt{2}\sqrt{\bar{S}AHT}.$$

Combining the above results, we have

$$\sum_{t=1}^{T}\mathbb{E}\left[\left(V^{\mathcal{M},\tilde{\pi}} - V^{\mathcal{M},\pi^t}\right)\right] = \sum_{t=1}^{T}\mathbb{E}\left[\left(V^{\mathcal{M}^t,\pi^t} - V^{\mathcal{M},\pi^t}\right)\right]$$

$$\le O\left(H\sqrt{M\log(AKH\tau_{\max}T)}\left[\tau_{\max}\bar{S}A + \sqrt{\bar{S}AHT}\right]\right)$$

$$= O\left(H^{\frac{3}{2}}\sqrt{M\bar{S}AT\log(AKH\tau_{\max}T)}\right)$$

$$= \tilde{O}\left(H^{\frac{3}{2}}\sqrt{M\bar{S}AT}\right)$$

$$\le \tilde{O}\left(H^{\frac{3}{2}}M\sqrt{KAT}\right). \tag{13}$$

Hence, we have proved the regret bound. **q.e.d.**

## B   Proofs for Propositions in Section 5

### B.1   Proof for Proposition 1

**Proof:** Let $V^*$ be the optimal value function of $\mathcal{M}$, and $V^*_{\mathcal{S}_G}$ be its restriction to $\mathcal{S}_G$. Let $\mathcal{V} \subset [0,H]^{|\mathcal{S}_G|}$ be the space of possible value functions $V^*_{\mathcal{S}_G}$. Note that by definition, $\mathcal{J}_i$ is the projection of $\mathcal{V}$ to $\mathcal{E}_i$, the exit states of $\mathcal{M}_i$.

Notice that $\mathcal{M}$ can be reduced to an MDP $\mathcal{M}_R$ with the same state space $\mathcal{S}_R = \mathcal{S}_G$ as the induced global MDP $\mathcal{M}_G$. For each state $s$ in $\mathcal{S}_R$, assume $s \in \mathcal{S}_i$, then its action space includes all the deterministic policies in subMDP $\mathcal{M}_i$. The transition and reward models $P^R$ and $r^R$ are defined similarly as $P^G$ and $r^G$. It is straightforward to see that an optimal policy in $\mathcal{M}_R$ perfectly recovers an optimal policy in $\mathcal{M}$. Let $\mathcal{T}$ be the dynamic programming operator in $\mathcal{M}_R$, and $\mathcal{T}'$ be the DP operator in $\mathcal{M}_G$, then we have

$$\mathcal{T}V - \Delta\mathbf{1} \le \mathcal{T}'V \le \mathcal{T}V \quad \forall V \in \mathcal{V},$$

where the first inequality follows from the definition of $\Delta$, and the second inequality follows from the fact that $\mathcal{T}$ has a larger action space.

We now prove Proposition 1 under Assumption 1 and a mild technical assumption that $\mathcal{T}^l\mathbf{0} \in \mathcal{V}$, for $l = 0, \ldots, |\mathcal{E}|$.

Let $L = |\mathcal{E}|$. Recall that $\mathcal{S}_G = \mathcal{E} \cup \{s_0\}$, thus $\mathcal{S}_G$ has at most $L+1$ states, and one of them is the terminal state $s_e$. Under Assumption 1, with VI with initial $V = \mathbf{0}$, both $\mathcal{T}$ and $\mathcal{T}'$ will compute the value function in $L$ iterations, that is $V^* = \mathcal{T}^L\mathbf{0}$, and $V^{\tilde{\pi}} = (\mathcal{T}')^L\mathbf{0}$. We now prove that $(\mathcal{T}')^l\mathbf{0} \ge \mathcal{T}^l\mathbf{0} - l\Delta\mathbf{1}$ for all $l = 0, 1, \ldots, L$ by induction. Notice that this inequality trivially holds for $l = 0$. Assume it holds for $l$, then we have

$$(\mathcal{T}')^{l+1}\mathbf{0} = \mathcal{T}'((\mathcal{T}')^l\mathbf{0}) \overset{(a)}{\ge} \mathcal{T}'(\mathcal{T}^l\mathbf{0} - \epsilon l\mathbf{1}) \overset{(b)}{=} \mathcal{T}'(\mathcal{T}^l\mathbf{0}) - \Delta l\mathbf{1}$$

$$\overset{(c)}{\ge} \mathcal{T}^{l+1}\mathbf{0} - \Delta\mathbf{1} - \Delta l\mathbf{1} = \mathcal{T}^{l+1}\mathbf{0} - \Delta(l+1)\mathbf{1},$$

where (a) follows from the induction hypothesis and the monotonicity of $\mathcal{T}'$, (b) follows from the "constant-shift" property of DP operator, and (c) follows from $\mathcal{T}^l \mathbf{0} \in \mathcal{V}$ by induction. Thus, we have $V^{\tilde{\pi}} = (\mathcal{T}')^L \mathbf{0} \geq \mathcal{T}^L \mathbf{0} - L\Delta \mathbf{1} = V^* - L\Delta \mathbf{1}$. So we have $V^{\tilde{\pi}}(s_0) \geq V^*(s_0) - L\Delta$.

Finally, we justify that the technical assumption $\mathcal{T}^l \mathbf{0} \in \mathcal{V}$, for $l = 0, \ldots, |\cup_i \mathcal{E}_i|$ is mild. Notice that we have $\mathbf{0} \leq \mathcal{T}\mathbf{0}$ since the rewards are non-negative. Thus, from the monotonicity of $\mathcal{T}$, we have

$$\mathbf{0} \leq \mathcal{T}\mathbf{0} \leq \mathcal{T}^2\mathbf{0} \leq \ldots \leq \mathcal{T}^L\mathbf{0} = V^*.$$

Define $\mathbb{V} = \{V : \mathcal{S}_G \to \Re^+ \text{ s.t. } 0 \leq V(s) \leq V^*(s) \, \forall s \in \mathcal{S}_G\}$. Thus, if $\mathbb{V} \subseteq \mathcal{V}$, then this technical assumption holds. **q.e.d.**

### B.2 Proof for Proposition 2

**Proof**: Recall that for any policy $\pi$, any exit value profile $J$ and any possible start state $s$, we have $V_J^\pi(s) = V_0^\pi(s) + \rho^\pi(s)J$, where $\rho^\pi(s)$ is a row vector encoding the probability distribution over the exit states when the start state is $s$ and policy $\pi$ is applied. Thus, for any exit values $J$ and $J'$, we have

$$V_J^\pi(s) = V_0^\pi(s) + \rho^\pi(s)J = V_0^\pi(s) + \rho^\pi(s)J' + \rho^\pi(s)[J - J']$$
$$= V_{J'}^\pi(s) + \rho^\pi(s)[J - J'] \leq V_{J'}^\pi(s) + \|J - J'\|_\infty,$$

where the last inequality follows from $\rho^\pi(s)[J - J'] \leq |\rho^\pi(s)[J - J']| \leq \|\rho^\pi(s)\|_1 \|J - J'\|_\infty = \|J - J'\|_\infty$.

Thus, if $\tilde{\mathcal{J}}_k$ is an $\epsilon$-cover for $\mathcal{J}_i$, then by definition, there exists $\tilde{J} \in \tilde{\mathcal{J}}_k$ s.t. $\|J - \tilde{J}\|_\infty \leq \epsilon$. So we have

$$V_J^*(s) = V_J^{\pi_J}(s) \overset{(a)}{\leq} V_{\tilde{J}}^{\pi_J}(s) + \epsilon \overset{(b)}{\leq} V_{\tilde{J}}^{\pi_{\tilde{J}}}(s) + \epsilon \overset{(c)}{\leq} V_J^{\pi_{\tilde{J}}}(s) + 2\epsilon, \tag{14}$$

where (a) and (c) follow from the inequality above and $\|J - \tilde{J}\|_\infty \leq \epsilon$, and (b) follows from that $\pi_{\tilde{J}}$ is an optimal policy with exit value $\tilde{J}$. Hence, we have $\Delta_i(\tilde{\mathcal{J}}_k) \leq 2\epsilon$. **q.e.d.**

### B.3 Proof for Proposition 3

**Proof:** Consider an arbitrary exit profile $J$ and an arbitrary start state $s$. Due to the deterministic exit assumption, under the deterministic optimal policy $\pi_J$, the agent will deterministically exit at an exit state $e \in J_i$.

One key observation is that under the policy $\pi_{J_e}$, the agent will also exit at $e$. To see it, notice that the fact that the agent exits at $e$ under $\pi_J$ implies that there exist policies under which the agent exits at $e$ from the start state $s$. Moreover, under $J_e$, for any deterministic policy $\pi$ that does not exit at $s_e$, we have $V_{J_e}^\pi(s) \leq H$. On the other hand, for any deterministic policy $\pi$ that exits at $e$, we have

$$V_{J_e}^\pi(s) = V_0^\pi(s) + H + 1 \geq H + 1.$$

Thus, $\pi_{J_e}$, the optimal policy under the exit value $J_e$, must exit at state $e$.

Hence we have:

$$V_J^*(s) \overset{(a)}{=} V_J^{\pi_J}(s) \overset{(b)}{=} V_{J_e}^{\pi_J}(s) + J(e) - J_e(e) \overset{(c)}{\leq} V_{J_e}^{\pi_{J_e}}(s) + J(e) - J_e(e)$$
$$\overset{(d)}{=} V_J^{\pi_{J_e}}(s) \leq V_J^*(s), \tag{15}$$

where (a) follows from the definition of $\pi_J$, (b) follows from the fact that under $\pi_J$, the agent exits at $e$, and (c) follows from the fact that $\pi_{J_e}$ is optimal under the exit value $J_e$, and (d) follows from the fact that under $\pi_{J_e}$, the agent exits at $e$. Consequently, $\pi_{J_e}$ is an optimal policy under the exit value $J$, and hence $\tilde{\mathcal{J}}_k = \{J_e : e \in \mathcal{E}_i\}$ satisfies $\Delta_i(\tilde{\mathcal{J}}_k) = 0$. **q.e.d.**