[Reviews · NeurIPS 2020]

Review 1

Summary and Contributions: The paper presents a novel framework for analyzing potential efficiencies in reinforcement learning due to hierarchical structure in MDPs. This framework formally defines several useful concepts (subMDPs, equivalent subMDPs, exit states and exit profiles) that allow for an elegant refinement of regret bounds in a well-defined regime.

Strengths: The identification of particular properties (subMDPs, exit state set, and equivalence of subMDPs) provides a clear and useful framework for theoretical analysis of hierarchical reinforcement learning. Overall this paper provides an elegant, concrete framework for formalizing hierarchical structure and quantifying the efficiency such structure may allow. As the authors discussing, this framework opens up further possibilities both in relaxing some of the assumptions and inspiring approaches to discovering exploitable structure.

Weaknesses: Tabular states, deterministic transitions, and an exact bijection for the subMDPs equivalent create a fairly restrictive set of assumptions. The authors address this directly and point to some promising avenues to relax these assumptions via established analysis of planning in approximate models. It will be especially fascinating to see if the assumptions can be relaxed on the bijection. As the motivating examples make clear, for real-world applications we do not have exact knowledge of the MDP or absolute certainty that the states and transitions of one subMDP ("rooms" and "floors" to the garbage collection robot) map exactly to each in an "equivalence class". However, the analysis provided here provides a useful starting point in that discussion. The lack of empirical evaluation is not really a weakness, as this paper provides a the essential theoretical groundwork on which to base a wide range of empirical evaluations and further theoretical analysis.

Correctness: The formulation is sound. The understanding of the general problem, assumptions used, and terms defined are clear, meaningful, and useful.

Clarity: The presentation of their specific formulation, relation to existing work, and theoretical analysis was clear and concise. The paper did an excellent job of situating this framework in established RL literature and pointing out the strong future potential along this line of work.

Relation to Prior Work: The analysis draws on and extends previous work in bottleneck states, option planning, and decomposing regret into planning and learning. It is well-situated in the existing literature and does a nice job of tying to established results, as well as recent developments, and pushing to novel insights.

Reproducibility: Yes

Additional Feedback: Line 54 isn't strictly accurate: in the general RL setting the agent does not necessarily know S (partial observability and function approximation) and often is not provided with an enumeration of which states are terminal (though most experiment frameworks provide a terminal flag as part of the observation). Nor would I say a fixed initial state (or set of s_0) is the most common default. None of this invalidates the analysis, but it would likely be better to rephrase so as not to imply unnecessary constraints or lack of awareness of the broader RL settings.


Review 2

Summary and Contributions: The paper presents theoretical results in hierarchical reinforcement learning. They show that if the problem can be broken into large number of small subproblems, then it can lead to large computational efficiency gains.

Strengths: The claims in the paper are well supported with theoretical analysis and builds on results from prior work. The assumptions of the claims and their practical implications are clearly explained.

Weaknesses: The paper assumes that the subproblems of an MDP are given to the agent, which is generally not true in practice. The paper does not address how such subproblems can be discovered automatically.

Correctness: Yes

Clarity: The paper is very well written and easy to follow. One minor comment: it will help if the paper explicitly mentions that is focusing on model-based RL algorithms.

Relation to Prior Work: Yes

Reproducibility: Yes

Additional Feedback: It will be interesting to consider the case where the subproblems are not precisely defined, and what would be resulting impact on the computational efficiency of the algorithm. Post Rebuttal: The authors have resolved my concerns in their response.


Review 3

Summary and Contributions: The paper provides a theoretical analysis of hierarchical reinforcement learning, deriving results on learning and planning efficiency when the reinforcement learning problem has repeated structure. The analysis is based on a decomposition of the base MDP into sub-MDPs using state partitions, capturing structure that is repeated exactly in multiple parts of the base MDP. There are two results. First, the authors extend an earlier regret bound by Osband et al (2013) and show the reduction in the bound possible through the repeated hierarchical structure in the MDP. This analysis is based on the algorithm PSRL (posterior sampling for reinforcement learning). Secondly, the authors analyze planning with options that are generated based on the repeated structure in the MDP. This analysis is based on Value Iteration and assumes that the state transition graph that corresponds to an optimal policy is acyclic. The authors provide a bound on the quality of the solution found based on the quality of the options (more specifically, the exit profiles that define the options). They also sufficient conditions for high-quality exit profiles.

Strengths: The paper formalises some of the benefits of hierarchical reinforcement learning, showing the precise impact of repeated structure on learning and planning efficiency. I found it useful and enjoyed reading the paper. The analysis can be a foundation for further work in the area, including new approaches to option discovery.

Weaknesses: The analysis is based on a decomposition of the MDP that looks for structure that is repeated exactly. An example is a navigation problem with a number of identical rooms. In future work, the authors discuss relaxing this assumption, as some other authors have done in the past. Rather than offering significant new insights into hierarchical reinforcement learning, the paper uses existing insights to define precisely the efficiencies that hierarchical reinforcement learning can introduce under some very basic settings and assumptions.

Correctness: I have not spotted any errors in the theoretical analysis (but I have not checked derivations carefully). There is no empirical analysis.

Clarity: Yes, the paper is generally clear and well organised.

Relation to Prior Work: There is a large literature on hierarchical reinforcement learning and model decomposition on MDPs. The authors adequately cite and discuss relevant pieces of this literature.

Reproducibility: Yes

Additional Feedback: The exact meaning of an exit profile is not clearly specified in the paper (section 5). The authors define it as a vector of values (definition 3) but does not clearly state what these values represent. The authors write: “To make this problem well defined, one needs to consider possible combinations of values associated with the exit states of a subMDP.” What are the “values” the authors refer to in this sentence? The authors continue: “For example, an agent which is in a room with two doors might consider making either door a subgoal, by giving it a high reward.” Does the exit profile specify the reward assigned to each exit state? The notion of exit profiles in the current paper is similar to exits defined by Hengst (ICML 2002), Discovering hierarchy in reinforcement learning with HEXQ. The authors may wish to indicate in the main paper that the proof for Theorem 1 is provided in Appendix 1 of the supplementary material. After rebuttal: The author response clarified the the exit profiles, thanks.


Review 4

Summary and Contributions: On Efficiency in Hierarchical Reinforcement Learning -------------------------------------------- This paper provides a theoretical justification for the properties of an MDP that would benefit from hierarchical decomposition. The justification relies on the Bayesian regret bounds to show that hierarchical decomposition can lead to statistically efficient learning by comparing the bounds for the “flat” MDP to the decomposed MDP, and thus deriving the following conditions for beneficial decomposition: either the subMDPs must all have a small state space or the original MDP is able to be decomposed into a small number of equivalent MDPs. The paper then goes on to discuss the computational complexity of planning with hierarchical structures with the Planning with Exit Profiles (PEP) algorithm. The authors derive a bound on the computational complexity of the PEP algorithm, which leads to the following properties being required for efficient learning: all subMDPs must be small, with a small number of exit profiles and total exit states. Finally, the paper also presents a bound on the performance of the PEP algorithm, and discusses the conditions for finding high-quality exit profiles, which is a requirement for the near-optimal performance of PEP.

Strengths: The paper is well organized, first introducing the metric for comparing the statistical efficiency of learning, the Bayesian regret bound. The bound is explained well, and it is clear how the bound is used to determine when decomposition would improve learning. The paper then discusses the computational complexity, which is the next logical step. The bounds for computational complexity make it clear how the requirements for computationally efficient planning were determined. I liked the line about amassing treasure. That's what it's all about!

Weaknesses: The main issue I had is that the paper does not clearly distinguish the connections to related work. In particular, I would like to know what the differences are to Mann et al's work, which the authors note in passing. How are these results connected to the "special cases"? Further, it seems these results rely heavily on previous work by Osband et al. It was not clear if these results (especially the first) was a straightforward extension. The acyclicity requirement is quite strong. As well, It was not clear if Prop 1 was using this requirement. To summarize, this theoretical paper is presenting several results on when HRL can be efficient. The results are quite intuitive and sensible. On the downside, there are questions about connections to prior work, and limited novel insight is developed.

Correctness: yes

Clarity: yes

Relation to Prior Work: no

Reproducibility: Yes

Additional Feedback: Thanks for the clarifications! The paper is acceptable.

[Author Response · NeurIPS 2020]

We thank all reviewers for spending their valuable time reviewing our paper. We appreciate their insightful comments,
and will incorporate them in the revision. We now answer some specific question in detail.

**Relaxation of the notion of equivalent subMDPs:** The definition of "equivalent subMDPs" (Definition 2) requires
that (1) there is a *bijection* between state spaces and (2) through which the subMDPs have the *same* transition/reward
models at internal states. As discussed in the paper, (2) can be relaxed to *similar* transition/reward models. The
bijection assumption (1) is essential for the planning algorithm (Algorithm 2) to work in its current form, and hence, for
the computational efficiency results. For the statistical efficiency results, this assumption could be relaxed, e.g. if a
clustering of the state space into clusters with similar rewards and transition distributions exists. We agree that finding a
different approach for Algorithm 2, not relying on the bijection, would be an interesting direction for future work.

**Automatically discovering subproblems:** We fully agree that how to automatically discover the subMDPs is an
important open problem. However, it is beyond the scope of this paper and we aim to address it in future work. The
results we provide are a necessary first step before tackling discovery.

**Connections to prior work:** We will add a more explicit discussion about the comparison to Mann et al. (2015)
to the paper. To summarize, they discuss and analyze two algorithms: *Option-Fitted Value Iteration (OFVI)* and
*Landmark-Approximate Value iteration (LAVI)*. The OFVI analysis relies on the discounted-average concentrability of
the future state distributions in the semi-MDP defined by options, so it is a very different-flavor result. LAVI relies on
options that go to designated landmark states, and which are computed by solving a deterministic relaxation of the
semi-MDP in a neighborhood of landmarks. In our terminology, such options have a single exit state, and LAVI then
solves the problem that jumps between landmarks. There is no repeating structure in this approach; in fact, each option
only applies in a small neighborhood of state space around a landmark. Our result could be applied to the LAVI setup
directly, but it would be hard to compare to their bound directly due to the very different quantities involved.

Theorem 1 in this paper is partially motivated by Osband et al. (2013); however, we consider a very different setting and
our result is technically more complex. Specifically, (1) Theorem 1 considers hierarchical structure while Osband et al.
(2013) does not; (2) Theorem 1 allows sub-optimal planning while Osband et al. (2013) does not; (3) Theorem 1 allows
a random time horizon $\tau$ while Osband et al. (2013) is restricted to a fixed time horizon. These major differences make
several key steps in the analysis both different and more challenging (e.g. the step to bound the probability that the
sampled MDP $\mathcal{M}^t$ is not in the confidence set $\mathbb{M}_t$ at each episode $t$) . We will further highlight these differences in the
revision.

**About the "Acyclicity" assumption:** We chose to make the "acyclicity" assumption (Assumption 1) mainly to simplify
the exposition of our computational efficiency results. This assumption ensures that value iteration (VI) will terminate
in a finite number of steps, a fact we use in Proposition 1. This assumption can be relaxed: by using the *weighted
sup-norm contraction* under VI in *stochastic shortest path problems* (see Section 3.3 of Bertsekas (2015)), we can
obtain a similar computational efficiency result without this assumption, but it is mathematically more complicated and
harder for readers to digest, and so we opted for the cleaner version. We will add an observation to the paper noting that
the "acyclicity" assumption is not strictly necessary, thanks for pointing that out!

**To Reviewer #1:** Assuming that (1) the agent knows $\mathcal{S}$ and (2) a fixed initial state is mainly to simplify the exposition.
We agree this does not represent the most encompassing formulation of RL, however, we believe that these simplifying
assumptions are without loss of generality and the insights obtained from this paper apply more broadly. We will further
clarify this point in the revision.

**To Reviewer #2:** We will clarify in the revision that this paper focuses on model-based RL algorithms.

It is not completely clear what is meant by "the case where the subproblems are not precisely defined". If you mean the
case where the subproblems need to be discovered, or where the current notion of equivalent subMDPs needs to be
relaxed, please see the discussion above.

**To Reviewer #3:** The exit profile of a subMDP can be any vector that assigns a real number to each exit state. Different
exit profiles will induce different policies in the subMDP. If an exit profile is close to (or equal to) the optimal state
values at the exit states, then it will induce a near-optimal (or optimal) policy. The notion of exit profiles in this paper
is different from *"exits"* defined by Hengst (ICML 2002). Specifically, the exit profiles in this paper are vectors (see
Definition 3), while the "exits" in Hengst (ICML 2002) are state-action pairs (see Definition 1 in their paper).

We will indicate in the main paper that the proof for Theorem 1 is provided in Appendix A of the supplementary
material. Thanks for the reminder!

**To Reviewer #4:** We believe that we have addressed all of your concerns above.

**To all reviewers:** Thanks again for reviewing our paper and reading this author response!

[Meta-Review · NeurIPS 2020]

Quoting from the reviewers: R1: The paper presents a novel framework for analyzing potential efficiencies in reinforcement learning due to hierarchical structure in MDPs. This framework formally defines several useful concepts (subMDPs, equivalent subMDPs, exit states and exit profiles) that allow for an elegant refinement of regret bounds in a well-defined regime. The identification of particular properties (subMDPs, exit state set, and equivalence of subMDPs) provides a clear and useful framework for theoretical analysis of hierarchical reinforcement learning. Overall this paper provides an elegant, concrete framework for formalizing hierarchical structure and quantifying the efficiency such structure may allow. The paper provides a theoretical analysis of hierarchical reinforcement learning, deriving results on learning and planning efficiency when the reinforcement learning problem has repeated structure. The analysis is based on a decomposition of the base MDP into sub-MDPs using state partitions, capturing structure that is repeated exactly in multiple parts of the base MDP. R3: There are two results. First, the authors extend an earlier regret bound by Osband et al (2013) and show the reduction in the bound possible through the repeated hierarchical structure in the MDP. This analysis is based on the algorithm PSRL (posterior sampling for reinforcement learning). Secondly, the authors analyze planning with options that are generated based on the repeated structure in the MDP. This analysis is based on Value Iteration and assumes that the state transition graph that corresponds to an optimal policy is acyclic. The authors provide a bound on the quality of the solution found based on the quality of the options (more specifically, the exit profiles that define the options). They also sufficient conditions for high-quality exit profiles. The paper formalises some of the benefits of hierarchical reinforcement learning, showing the precise impact of repeated structure on learning and planning efficiency. I found it useful and enjoyed reading the paper. The analysis can be a foundation for further work in the area, including new approaches to option discovery.